# Terra: Imperative-Symbolic Co-Execution of Imperative Deep Learning Programs

**Taebum Kim**
Seoul National University, FriendliAI
k.taebum@snu.ac.kr,
ktaebum@friendli.ai

**Eunji Jeong**[*]
Samsung Research
eun-ji.jeong@samsung.com

**Geon-Woo Kim**
Seoul National University, FriendliAI
gwsshs22@snu.ac.kr,
gwsshs22@friendli.ai

**Yunmo Koo**
Seoul National University, FriendliAI
mpbb03@snu.ac.kr,
yunmorning@friendli.ai

**Sehoon Kim**[*]
University of California, Berkeley
sehoonkim@berkeley.edu

**Gyeong-In Yu**
Seoul National University
gyeongin@snu.ac.kr

**Byung-Gon Chun**[†]
Seoul National University, FriendliAI
bgchun@snu.ac.kr,
bgchun@friendli.ai

## Abstract

Imperative programming allows users to implement their deep neural networks (DNNs) easily and has become an essential part of recent deep learning (DL) frameworks. Recently, several systems have been proposed to combine the usability of imperative programming with the optimized performance of symbolic graph execution. Such systems convert imperative Python DL programs to optimized symbolic graphs and execute them. However, they cannot fully support the usability of imperative programming. For example, if an imperative DL program contains a Python feature with no corresponding symbolic representation (e.g., third-party library calls or unsupported dynamic control flows) they fail to execute the program. To overcome this limitation, we propose Terra, an imperative-symbolic co-execution system that can handle any imperative DL programs while achieving the optimized performance of symbolic graph execution. To achieve this, Terra builds a symbolic graph by decoupling DL operations from Python features. Then, Terra conducts the imperative execution to support all Python features, while delegating the decoupled operations to the symbolic execution. We evaluated Terra's performance improvement and coverage with ten imperative DL programs for several DNN architectures. The results show that Terra can speed up the execution of all ten imperative DL programs, whereas AutoGraph, one of the state-of-the-art systems, fails to execute five of them.

---

[*]Work done at Seoul National University
[†]Corresponding author

35th Conference on Neural Information Processing Systems (NeurIPS 2021).

# 1 Introduction

The rapid evolution of deep neural networks (DNNs) has been fueled by the support of deep learning (DL) frameworks [6, 32]. Such frameworks provide users with a programming layer to build and execute DNNs, commonly adopting Python as their host language. Typically, they execute DL programs with one of the two execution models: imperative or symbolic execution. In the former, the Python interpreter executes a DL program as a normal program, invoking DL operations on-the-fly. The invoked DL operations are executed on a separate DL accelerator asynchronously, and the Python interpreter continues running the program. The dynamic control flows of the DL operations are naturally expressed by the interpretation of the program, and users can utilize any functionalities of Python (e.g., dynamic typing and third-party libraries [9, 16]) while executing DL operations. On the other hand, in the latter model, the Python interpreter embeds DL operations into a symbolic graph that represents the entire dataflow of a DNN. Thus, users should define their DL programs only with existing symbolic operations that DL frameworks support. In other words, the dynamic control flows of a DNN should be explicitly represented by control flow operations (e.g., `tf.cond` and `tf.while` of TensorFlow). However, the symbolic execution can take advantages of various optimization techniques because the symbolic graph contains a whole computation lineage of a DNN architecture.

Although symbolic execution achieves higher performance compared to imperative execution, imperative execution has been preferred because of its usability. Several systems [2, 3, 4, 15, 19, 29, 39] have been proposed to match the speed of symbolic execution while enjoying the benefit of imperative execution. These systems attempt to generate a symbolic graph that represents an entire imperative program and execute the graph instead of imperatively running the program. Methods for generating the symbolic graph can be broadly classified into two approaches: *single path tracing* and *static compilation*. The former approach generates a symbolic graph by imperatively executing a single iteration of a program and recording the executed DL operations. Systems that adopt the latter approach translate the abstract syntax tree (AST) of a program into a symbolic graph.

Unfortunately, both approaches can correctly handle only a subset of imperative DL programs. For example, dynamic control flows in an imperative program are not captured by the *single path tracing* approach. On the other hand, the *static compilation* approach cannot correctly generate a symbolic graph if a target program contains an AST node that does not have a corresponding symbolic operation such as *try-except*s, *generator*s, *Python object mutation*s, and *third-party library call*s. As a result, it is up to the users to clearly understand the limited usability of these systems.

In this paper, we propose Terra, an imperative-symbolic co-execution system that addresses the limitations. While the previous approaches replace the imperative execution with the symbolic execution, Terra maintains the imperative execution to support all Python features where DL operations are delegated to the symbolic execution. Also, Terra generates a symbolic graph by utilizing multiple traces of an imperative program. To be specific, Terra imperatively runs an imperative DL program for several iterations and collects traces, each of which is a linear chain of DL operations sorted by the execution order. The collected traces are merged into a TraceGraph, a directed acyclic graph (DAG) that encapsulates the captured DL operations along with their diverse execution orders. Terra stops collecting traces when the trace of the latest iteration is already embedded in the TraceGraph. To generate an executable symbolic graph from the TraceGraph, Terra adds additional information to the TraceGraph. First of all, Terra annotates the TraceGraph to enable communication between the imperative execution and the symbolic execution. In addition, Terra further analyzes the TraceGraph to insert control flow operations explicitly, so that a symbolic graph can be executed with the DL operations in a certain trace. After generating a symbolic graph from the TraceGraph, Terra starts the co-execution of a skeleton imperative program, in which DL operations are not performed, and the symbolic graph that represents the DL operations. Here, Terra continually checks whether the current trace is being expressed by the TraceGraph. If Terra detects a new trace, Terra discards the symbolic graph and collects more traces by running the original program imperatively. Terra then re-generates the symbolic graph and restarts the co-execution. Consequently, Terra is able to run any imperative DL programs correctly and efficiently even if it contains the Python features that the previous approaches cannot handle.

We have implemented Terra on TensorFlow v2.4.1 and compared Terra with TensorFlow's imperative execution [7] and AutoGraph [29]. Our evaluation shows that Terra can train ten imperative DL programs including convolutional neural networks, transformer-based networks, and generative

```
1  def train_step(x):              1  def generator():                1  dr = Dropper()
2    y = library_call1()           2    for _ in tf.range(N):         2  dr.drop_prob = 0.0
3    loss = model(x, y)            3      yield tf.random.normal()    3  def train_step(step, x):
4    library_call2(                4                                  4    if step > 100:
5      loss.numpy())               5  def train_step(x):            5      dr.drop_prob = 0.8
                                   6    for y in generator():         6    x = tf.nn.dropout(
                                   7      x = x + y                   7      x, dr.drop_prob)
                                   8    return x                      8    return x
```

(a) third-party library call & tensor materialization
(b) dynamic control flow
(c) Python object mutation

Figure 1: Simple examples that the *static compilation* approach cannot deal with. Note that AutoGraph could silently produce an incorrect result in the Python object mutation case.

adversarial networks up to 1.73x faster than the original imperative execution. However, AutoGraph fails to support five programs for three reasons: third-party library call, Python object mutation, and tensor materialization during conversion, which we describe in § 2.2.

## 2   Background & Related Works

### 2.1   Imperative and Symbolic Execution

The **imperative execution** (a.k.a. define-by-run) [30, 32, 43] treats a DL program entirely as a typical Python program. Whenever the Python interpreter encounters a statement that declares a DL operation (e.g., z = torch.matmul(x, y)), the interpreter asynchronously invokes a corresponding computation kernel of a DL accelerator (typically GPU or TPU). The imperative execution highly improves the programmability of DL programs because users can fully utilize the convenient language features and rich ecosystem of Python, including built-in functions, dynamic control flows, dynamic typing, and third-party libraries. However, since the imperative execution cannot obtain a whole view of DNN computation, it misses optimization opportunities that the symbolic execution explained below can carry out.

The **symbolic execution** (a.k.a. define-and-run) [6, 14, 42] executes a pre-built symbolic graph. In the symbolic execution, users should express their DNN architectures as symbolic graphs using the three kinds of symbolic operations: DL operations, control flow operations, and auxiliary operations. The DL operations are conventional compute-intensive operations (e.g., matrix multiplication or convolutional layer), and the control flow operations determine which DL operations to be executed based on a tensor value within the graph. Finally, to mitigate the limited usability of the symbolic execution, it supports auxiliary operations (e.g., tf.print and tf.py_function of TensorFlow). After the symbolic graph is constructed, graph optimizations could be applied such as operation fusion [11, 20, 26, 31, 35, 41, 46, 47, 48], parallelized execution [23, 31, 35], device placement [27, 28, 48], layout optimization [20, 25], and memory optimization [8, 17, 18]. Then, an optimized graph executor such as TVM [11], TensorRT [31], TFRT [40], and XLA [41] undertakes the actual execution of the symbolic graph.

### 2.2   Imperative Program with Symbolic Execution

There are two approaches to gain the usability of imperative execution and the performance of symbolic execution simultaneously. They attempt to convert an imperative DL program to a symbolic graph and exploit the symbolic execution with the converted graph. The **single path tracing** approach (e.g., torch.trace [3], JAX [15], and tf.function [39]) executes an imperative DL program once and records all DL operations that were executed. A single linear chain of the executed DL operations, which is called a trace, becomes a symbolic graph of the imperative program. The symbolic graph is executed instead of the imperative program for subsequent iterations. Although the *single path tracing* approach looks very simple and intuitive, it is hard to capture dynamic control flows of the imperative program with this approach. To reflect them correctly, users need to explicitly declare the control flow operations for all dynamic control flows in their imperative programs, which undermines the programmability of the imperative program. Moreover, Python features that do not have corresponding symbolic operations (e.g., mutation of Python objects, use of third-party libraries)

are neither captured by the trace. It can yield an incorrect result since, in the following iterations, a graph executor executes a symbolic graph, which does not contain the Python features.

The **static compilation** approach (e.g., TorchScript [4] and JANUS [19]) is proposed to resolve the problem of the *single path tracing* approach. In contrast to the *single path tracing* approach, the *static compilation* approach does not extract a trace to generate a symbolic graph. Instead, it traverses an abstract syntax tree (AST) of the imperative program and directly converts each AST node to a corresponding symbolic operation. Even if it guarantees the correctness of the program, it fails to convert a program to a symbolic graph if the program contains a Python feature with no corresponding symbolic operation. We classify the representative failure cases into four categories: *third-party library call*, *tensor materialization during conversion*, *dynamic control flow*, and *Python object mutation*. Figure 1 presents simple examples of the cases. The *static compilation* approach cannot convert the third-party library calls of Figure 1a and fails when it attempts to materialize the tensor data (`loss.numpy()`) during the conversion. Figure 1b shows the `generator` in Python that the *static compilation* approach cannot convert. For Figure 1c, only JANUS handles it by implementing custom auxiliary operations (i.e., *GetAttr* and *SetAttr* operations) that access the Python heap during the symbolic execution.

To increase the practicality of the previous approaches, AutoGraph [29] combines the *static compilation* approach with the *single path tracing* approach. AutoGraph converts AST nodes that correspond to dynamic control flow such as *if-else*, *for*, and *while* to new AST nodes representing proper control flow operations such as `tf.cond` and `tf.while`. Then, AutoGraph generates a symbolic graph by applying the *single path tracing* approach to the converted AST. Unfortunately, it cannot fully support various kinds of dynamic control flows such as *generator* and *try-except* of Python. Hence, AutoGraph also entails the same correctness problem of the *single path tracing* approach when the imperative DL program employs the features described in Figure 1.

When the *static compilation* approach detects unsupported features, it just raises an error and aborts the program execution. To avoid this, users have to write their programs using only the supported features or put in additional efforts such as annotating types and refactoring functions to use tensor objects for their inputs and outputs. In this regard, AutoGraph [29] and TorchScript [4] provide the official language references [1, 5] that users should be aware of before writing a target imperative program. However, those references usually require expert knowledge to understand, imposing a steep learning curve for new users. For example, the language document of AutoGraph [1] spans roughly twelve pages and divides features that AutoGraph does not support into four categories and ten subcategories. TorchScript provides an official language specification [5] by enumerating supported features over eleven pages.

LazyTensor [36] is a concurrent work related to ours. It uses the *lazy evaluation* approach, in which the Python interpreter and symbolic graph executor run alternately. The Python interpreter executes an imperative DL program as it is and extracts a linear trace of operations. LazyTensor then checks whether the extracted trace is already cached or not. If cached, LazyTensor directly executes the cached graph. If not, it compiles and executes the new graph, then stores the graph for further executions. With the lazy evaluation, LazyTensor could support all Python features while executing the symbolic graph as Terra does. However, it has an unavoidable performance overhead due to the alternate execution. In other words, the graph executor progresses only when the Python interpreter finishes checking the existence of the cached graph. Similarly, the Python interpreter progresses after finishing the graph execution. Moreover, if an imperative program has a dynamic control flow that is not determined by a tensor value, LazyTensor fails to capture the control flow transparently. In this case, LazyTensor would work inefficiently because the traces could be different for each iteration.

## 3 Our Approach: Imperative-Symbolic Co-Execution

To the best of our knowledge, no existing DL framework can completely convert an arbitrary imperative DL program into a symbolic graph. We believe that a one-to-one mapping from all Python features to corresponding symbolic operations should exist to support all imperative programs with the approaches. In other words, building such a mapping is the same as covering all Python syntax with symbolic operations. Moreover, the mapping should be updated as a new feature of Python (e.g., pattern matching of Python 3.10 [33]) is introduced. Eventually, it is equal to building a new

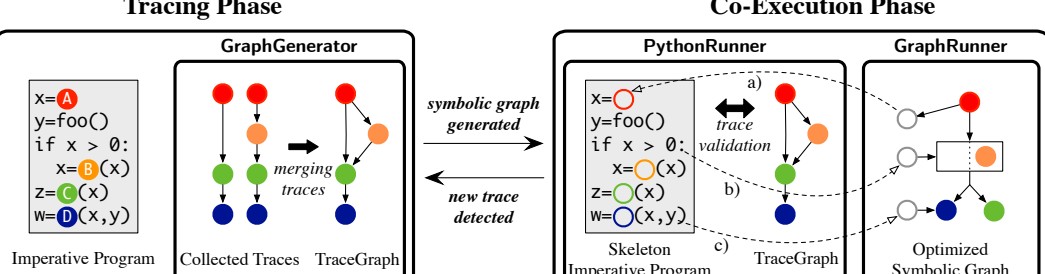

Figure 2: An overview of Terra. Each dotted arrow denotes a) the PythonRunner fetches a tensor value from the GraphRunner, b) the PythonRunner informs the GraphRunner of the path that the PythonRunner takes, and c) the PythonRunner feeds an external tensor to the GraphRunner. Rectangle in the optimized symbolic graph denotes the control flow operation.

Python execution engine for a symbolic graph representing a Python program itself, which requires a tremendous amount of time and effort.

Therefore, we take a different approach that does not replace the entire imperative execution with the symbolic execution as the previous approaches do. Instead, we let the Python interpreter run an imperative program to support all Python features naturally while separating only DL operations from the imperative execution. As the Python interpreter executes the program except performing DL operations, the decoupled DL operations are executed by a graph executor simultaneously. To enable the co-execution, we generate a symbolic graph representing DL operations that would have been launched by the Python interpreter. While the previous approaches have to build a complete symbolic graph that encapsulates all semantics of the DL program for correctness, we construct a symbolic graph solely based on collected traces of DL operations. Although we do not embed all semantics of the DL program in the symbolic graph, it can handle any DL program by the co-execution of the skeleton program complementing the symbolic execution. Within the skeleton program, DL operations are not performed but all other Python features are preserved as the original imperative program. Executing the symbolic graph in parallel with the skeleton program, we fully achieve the usability of the imperative execution along with the optimized performance of the symbolic execution.

## 4 System Design

Terra is a system that realizes our imperative-symbolic co-execution approach. In this section, we describe how Terra implements the co-execution of a skeleton imperative program and a symbolic graph in detail. There are two requirements to seamlessly maintain the imperative execution along with executing the symbolic graph. First of all, Terra should allow exchanging tensor values between the imperative execution and the symbolic execution when there exists data dependency between each other (e.g., Figure 1a). Furthermore, the imperative execution should inform the symbolic execution of the correct choice of path to follow because the execution flow of the program is determined by the Python interpreter. To achieve these, Terra implements new symbolic operations for such communication and inserts them into the symbolic graph. In the following, we first describe the entire process of the imperative-symbolic co-execution (§ 4.1). Next, we explain how Terra merges collected traces into the TraceGraph and generates a symbolic graph from it in detail (§ 4.2).

### 4.1 Imperative-Symbolic Co-Execution

The co-execution of Terra consists of the following two phases: the *tracing phase* and the *co-execution phase*. Terra begins execution in the tracing phase, as shown in Figure 2. In this phase, the conventional imperative execution is carried out with the given imperative DL program. At the same time, the GraphGenerator collects traces of each iteration. The GraphGenerator incrementally merges the traces into the *TraceGraph*, a directed acyclic graph (DAG) that encapsulates all the collected traces. Since the number of possible traces during the imperative execution cannot be determined, the GraphGenerator collects traces until the trace of the latest iteration is fully covered in the TraceGraph. In such a case, the GraphGenerator generates a symbolic graph from the TraceGraph.

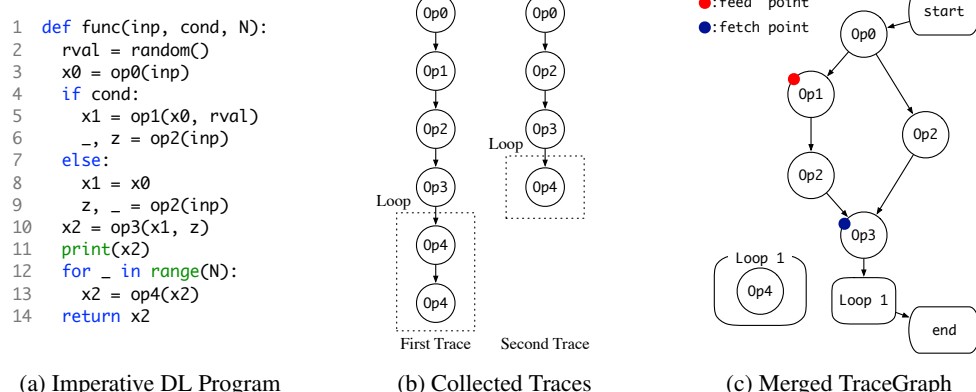

```
1  def func(inp, cond, N):
2      rval = random()
3      x0 = op0(inp)
4      if cond:
5          x1 = op1(x0, rval)
6          _, z = op2(inp)
7      else:
8          x1 = x0
9          z, _ = op2(inp)
10     x2 = op3(x1, z)
11     print(x2)
12     for _ in range(N):
13         x2 = op4(x2)
14     return x2
```

(a) Imperative DL Program          (b) Collected Traces          (c) Merged TraceGraph

Figure 3: Illustration of how the TraceGraph is merged from the imperative DL program.

With the generated symbolic graph, Terra enters the co-execution phase. In this phase, Terra uses the PythonRunner and the GraphRunner. The PythonRunner executes a *skeleton imperative program* that does not launch DL operations anymore. The GraphRunner executes the generated symbolic graph with a separate graph executor. For each DL operation, the PythonRunner skips the actual computation and creates an empty tensor object(s) as an output(s) of the operation. If the Python-Runner has to materialize an empty tensor (e.g., print a loss value), it fetches the actual data from the GraphRunner. Similarly, the GraphRunner might need an external tensor (e.g., an input data, a Python primitive value) from the PythonRunner. Terra implements new symbolic operations to establish the communication. For each communication, a Runner that needs the data from the other waits until the required data becomes ready.

For every iteration in the co-execution phase, the PythonRunner keeps a trace being made by the DL operations in the current iteration. The PythonRunner continuously compares the trace with the TraceGraph to notify the GraphRunner of the current control flow and check the validity of the symbolic graph in the GraphRunner. If the latest DL operation in the trace indicates that the PythonRunner takes a specific path, it informs the GraphRunner of the path with a new symbolic operation, which sets a conditional input of a corresponding control flow operation in the symbolic graph. For example, if the PythonRunner takes the true path of the skeleton imperative program of Figure 2 (i.e., if x > 0:), the GraphRunner receives such information from the PythonRunner and executes the operation of the true path. Furthermore, if the latest DL operation is not expressed in the TraceGraph, Terra considers the current trace as a new trace that the existing symbolic graph cannot handle. Terra then cancels the execution of the GraphRunner and falls back to the tracing phase. Thereafter, the GraphGenerator collects more traces and generates a new symbolic graph covering more traces than before to continue the co-execution.

### 4.2 Symbolic Graph Generation

In this section, we describe how the GraphGenerator merges the collected traces into the TraceGraph and then generates a symbolic graph from the TraceGraph.

**TraceGraph**. Each node of the TraceGraph corresponds to a DL operation, and each edge denotes an execution order between two nodes. For example, if a *Conv2D* operation is followed by another *ReLU* operation in a single trace, the TraceGraph has a directed edge from a *Conv2D* node to a *ReLU* node. For the first trace, the TraceGraph contains a single linear chain of nodes that have two extra nodes; the *start* node and the *end* node. Those nodes do not correspond to DL operations but for indicating the start point and the end point of the merging.

For subsequent traces, the GraphGenerator attempts to match each operation of the trace with an existing node of the TraceGraph. The GraphGenerator uses a pointer that points to the latest matched node of the TraceGraph, which initially points to the *start* node. For each operation, the GraphGenerator checks whether there exists an equal node among the children of the latest matched node. To check the equality, the GraphGenerator compares the type of operation (e.g., Conv2D and MatMul), attributes of operation (e.g., a filter size and a kernel size of convolution), and whether two operations were executed at the same location of the program. If the GraphGenerator finds the

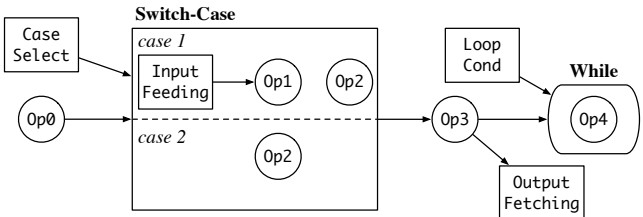

Figure 4: Generated symbolic graph from the TraceGraph of Figure 3c

child node of the latest matched node which satisfies all criteria, the GraphGenerator updates the latest matched node to that child node, not creating a new node. It then continues merging the next operation of the trace. If all the operations are matched, the GraphGenerator sets the latest matched node to the *end* node, denoting that the current trace is already captured by the TraceGraph. The definition of the equality criteria is in Appendix A.

When the GraphGenerator fails to match the operation with the existing nodes, it denotes that a new trace is detected. A new node is created by creating a new branch from the latest matched node. While expanding the TraceGraph for the new trace, the expanded branch could be merged back into the pre-existing branch if there is a node that is not a child of the latest matched node but satisfies all criteria of equality. For example, Figure 3c depicts the TraceGraph built from the program of Figure 3a, which first took the *true* path (line 5-6) and took the *false* path (line 8-9) at the second execution. From the program, the GraphGenerator collects two different traces as shown in Figure 3b. When the GraphGenerator attempts to merge the second trace into the TraceGraph that contains the first trace, *Op2* of the second trace cannot be matched with *Op1* and cannot be merged back into *Op2* of the first trace because two *Op2*s were executed in different locations. Thus, the node for *Op2* is created in the right branch of Figure 3c, and the branch is merged when the GraphGenerator succeeds to match *Op3*.

As shown in Figure 3c, the GraphGenerator merges the nodes that are executed in the same loop of the program. The GraphGenerator is aware of the loop because it compares the program location where DL operations were executed. It then groups those nodes within an extra loop node and conducts merging the nodes separately. For example, *Loop 1* in Figure 3c is the loop node for the loop of Figure 3a (line 12-13). The GraphGenerator merges the second *Op4* of the first trace with the first *Op4* of the first trace because they were executed in the same loop. Also, *Op4* of the second trace is merged to the same node.

**Communication Point**. To create the symbolic operations for data communication between the PythonRunner and the GraphRunner, the GraphGenerator captures communication points and annotates such points in the TraceGraph. Those points are classified into *feed* points and *fetch* points. The feed point is where the operation gets an input from the Python interpreter such as training data and Python primitive values. Similarly, the fetch point is where the Python interpreter needs a value of the DL tensor. For example, *Op1* in Figure 3a receives *rval* as an input (line 5), and the Python interpreter needs the value of *x2* (line 11) to print it out. The GraphGenerator captures those points and annotates them in the corresponding nodes of the TraceGraph.

**Symbolic Graph Generation**. The GraphGenerator converts the nodes in the TraceGraph to the corresponding DL operations and creates additional *Input Feeding* and *Output Fetching* operations to establish data communication during the co-execution. The *Input Feeding* operation corresponds to the feed point of the TraceGraph, enabling the PythonRunner to feed an external tensor to the GraphRunner. Similarly, the *Output Fetching* operation corresponds to the fetch point of the TraceGraph, allowing the PythonRunner to fetch materialized DL tensor from the GraphRunner. As a result, the GraphGenerator represents the entire computation lineage in the single graph with the communication operations. Without those operations, the GraphGenerator should split the symbolic graph into smaller subgraphs at every feed-fetch point, which cannot efficiently apply additional optimizations.

To handle the diverse control flows in the TraceGraph, GraphGenerator utilizes the *Switch-Case* operation (e.g., `tf.case` of TensorFlow), which allows executing only a single *case* that depends on a particular condition. For the conditional input that informs the *Switch-Case* operation of which *case* to execute, the GraphGenerator creates the *Case Select* operation along with the *Switch-Case* operation,

| Program | Reason of the failure |
|---|---|
| DropBlock [12] | Python object mutation |
| MusicTransformer [21] | Python object mutation |
| SDPoint [22] | Python object mutation |
| BERT-CLS [24] | third-party library call |
| FasterRCNN [44] | tensor materialization during conversion |

Table 1: The programs that AutoGraph fails to execute and the reason for the failures. Note that Terra can execute all of them.

as shown in Figure 4. When the PythonRunner takes a certain path, it notifies the GraphRunner via the *Case Select* operation. Here, the GraphGenerator uses our *case assignment algorithm* that takes the TraceGraph as an input, traverses the TraceGraph, and returns the *Switch-Case* operations representing the control flows correctly. The formal description of the algorithm and the proof that the algorithm can handle any DAGs are described in Appendix B.

Finally, the GraphGenerator creates the *While* operation (e.g., `tf.while` of TensorFlow) for a loop node of the TraceGraph. As the *Case Select* operation, the GraphGenerator creates the *Loop Cond* operation along with the *While* operation. The PythonRunner informs the GraphRunner of whether the PythonRunner goes to the next iteration of the loop or exits the loop via the *Loop Cond* operation. As an optimization, the GraphGenerator unrolls the *While* operation if the loop node took the same number of iterations in the collected traces.

## 5 Evaluation

In this section, we evaluate Terra in the following two aspects:

- Can Terra exploit symbolic execution from imperative DL programs that AutoGraph, the static-compilation-and-tracing approach, cannot? (§ 5.2)
- How much does Terra speed up imperative DL programs? (§ 5.3)

### 5.1 Implementation and Experiment Setup

**Frameworks**. We use TensorFlow [6] v2.4.1 as our baseline DL framework. We have built Terra on TensorFlow v2.4.1, and our approach is applicable to other DL frameworks if they support both imperative and symbolic execution (e.g., MXNet [10] and PyTorch [32]). More details about the implementation of Terra are described in Appendix C. All evaluated imperative DL programs are implemented with the imperative API of TensorFlow, which has become the standard interface since TensorFlow v2.0. We compare Terra with TensorFlow imperative execution and with Auto-Graph [29], a state-of-the-art system that combines the *static compilation* approach with the *single path tracing* approach. We compile a single training step function of each imperative DL program (i.e., `@tf.function(autograph=True)`).

**Environments**. We conduct all the experiments on a single machine that is equipped with 8-core AMD Ryzen 7 2700X @ 3.7GHz and an NVIDIA TITAN Xp GPU. We use Ubuntu 18.04, CUDA 11.0, cuDNN 8.0, and Python 3.8.8.

**Imperative DL Programs**. For the experiments, we use ten imperative DL programs collected from open-source GitHub repositories: DropBlock [12], BERT-Q&A [13], MusicTransformer [21], SDPoint [22], BERT-CLS [24], GPT2 [34], DCGAN [37], ResNet50 [38], FasterRCNN [44], and YOLOv3 [45].

### 5.2 Imperative Program Coverage

Terra handles all the benchmark programs successfully with the imperative-symbolic co-execution. However, since AutoGraph does not support the entire set of Python features, it fails to execute five out of ten programs. Table 1 shows why AutoGraph fails to handle the programs and the detailed code snippets are described in Appendix D.

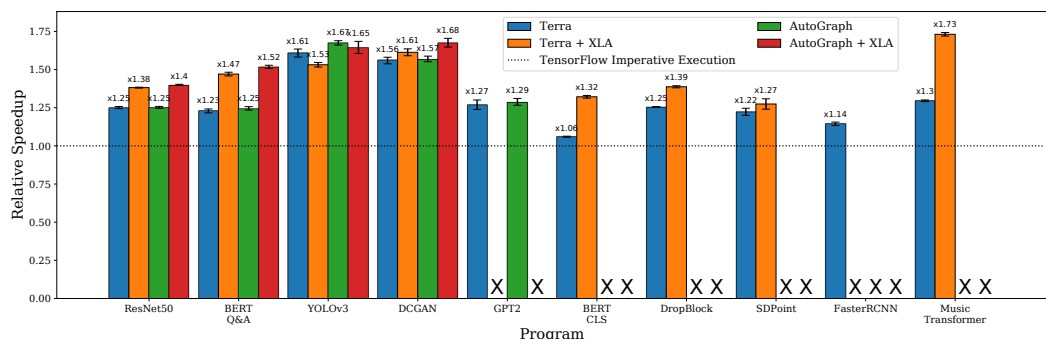

Figure 5: The training speed-up results of Terra, AutoGraph, and when applying XLA [41] to both systems relative to TensorFlow imperative execution. The dotted line presents the training throughput of the imperative execution. Note that Terra and AutoGraph share the same upper bound of performance improvement from the symbolic execution of TensorFlow.

According to the AutoGraph language document [1], more failures could exist if an imperative DL program contains unsupported Python features such as the use of Python *generator*, *try-except*, and *None* type values. To resolve all the limitations in AutoGraph, new functions to handle each failure should be implemented and it requires a huge engineering effort. Terra simply avoids the conversion-related problems by the imperative-symbolic co-execution.

### 5.3   Training Throughput

Figure 5 presents the training speed-up of Terra compared to TensorFlow imperative execution. For all programs, we measure the average training throughputs from 100 to 200 steps, and each experiment is conducted ten times. Terra achieves higher performance than the imperative execution for every program. To estimate whether Terra fully achieves the optimized performance of symbolic execution, we also compare the performance of Terra with AutoGraph, which shares the same graph executor of TensorFlow with Terra. AutoGraph closely follows the performance of the symbolic execution because it totally replaces the imperative execution with the symbolic execution. For the five programs that AutoGraph can execute, the performance improvements of Terra are on par with AutoGraph, which shows that Terra highly achieves the symbolic execution's optimized performance. Experiment settings such as batch size and the dataset are included in Appendix E.

Since Terra generates a symbolic graph and utilizes the symbolic execution, we evaluate the performance of Terra by applying XLA [41] as shown in Figure 5. Compared to the imperative execution, Terra improves the performance of seven programs by up to 1.73x when applying XLA. XLA is not applicable to GPT2 and FasterRCNN due to the dynamic shape of the input data. For each training iteration, the shapes of input data to the models can change. However, XLA cannot efficiently handle dynamic shapes because it assumes static shapes. For YOLOv3, we profile the execution and find that the current XLA fails to efficiently cluster operations for YOLOv3. To be more specific, YOLOv3 includes some DL operations such as *ResizeNearestNeighbor* and *Where*, which are not supported by XLA. Thus, XLA cannot efficiently fuse DL operations. In addition, we observe that Terra's performance decreases more than that of AutoGraph for YOLOv3 because the schedules of some *Output Fetching* operations are reordered because of XLA kernels, causing a longer stall in the PythonRunner. However, this problem can be addressed by extending XLA to support our custom operations.

We profile the execution of both PythonRunner and GraphRunner to analyze the performance of Terra. We focus on the performance analysis in the *co-execution* phase because Terra's execution is mostly in this phase. The number of transitions between the two phases and the overhead analysis for the *tracing* phase are included in Appendix F. Figure 6 shows the performance breakdown of the two Runners in a single training step. 'PythonRunner Exec' denotes the Python interpreter's active running time, such as executing user code or validating the symbolic graph. Both 'PythonRunner Stall' and 'GraphRunner Stall' indicate the stall time in which the PythonRunner waits for the GraphRunner to fetch the materialized tensor or vice versa. Finally, we measure GPU's active time to run the CUDA kernels along with the overhead of TensorFlow's graph executor as 'GraphRunner Exec'. For all

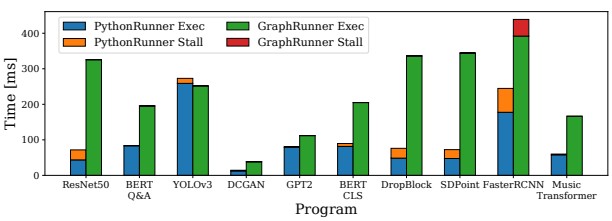

Figure 6: Performance breakdown within a single training step for both the PythonRunner and the GraphRunner.

| Program | Terra | Terra LazyEval |
|---------|-------|----------------|
| ResNet50 | x1.25 | x1.13 |
| BERT Q&A | x1.23 | x0.94 |
| DCGAN | x1.56 | x1.34 |

Table 2: Comparison of the training speed-up between Terra and Terra with lazy evaluation. The results are relative speed-up to TensorFlow imperative execution as Figure 5.

programs except for FasterRCNN, the GraphRunner is not stalled, implying that the GraphRunner fully exploits the optimized performance of the symbolic execution. In FasterRCNN, the stall of the GraphRunner occurs when the PythonRunner receives a materialized tensor from the GraphRunner and feeds it back to the GraphRunner. For YOLOv3, the PythonRunner's execution time is longer than that of the GraphRunner, which yields the slightly larger performance gap between Terra and AutoGraph in Figure 5.

Moreover, Figure 6 shows that the GraphRunner takes a longer time than the PythonRunner in most cases. The result implies the reason why the performance improvements of Terra are comparable to the performance improvements of AutoGraph. The execution of the PythonRunner is efficiently concealed by the execution of the GraphRunner with the co-execution. To demonstrate the effect of the co-execution, we serialize the execution of the PythonRunner and the GraphRunner then evaluate the performance for the simple programs among our benchmarks. Within the serialized execution, the GraphRunner does not start the execution along with the PythonRunner. Instead, it starts the execution when the PythonRunner requires tensor data through the *Output Fetching* operation. Eventually, the serialized execution is the same as the lazy evaluation that LazyTensor [36] does. Our results in Table 2 show that the lazy evaluation cannot fully achieve high performance of the symbolic execution. Even worse, it could become slower than the imperative execution when the execution time of the GraphRunner is not much longer than the execution time of the PythonRunner.

## 6 Conclusion

We propose Terra, a novel approach to execute imperative Python DL programs. Terra performs imperative-symbolic co-execution, which addresses the problem of converting an imperative program to a symbolic graph completely. Terra generates a symbolic graph only from the DL operations of an imperative DL program. It then carries out the imperative execution, simultaneously executing the symbolic graph. Therefore, Terra achieves optimized performance while maintaining all Python features of the imperative program. Our evaluation shows that Terra can speed up all imperative DL programs, even for the programs that AutoGraph cannot handle.

## Broader Impact

Our work aims to accelerate the execution of imperative DL programs while maintaining programmability. Our work is not associated with a specific application because our approach is applicable for any imperative DL programs. Thus, we believe our work does not have a significant impact on any audience from either an ethical or societal perspective, at the application level.

## Acknowledgements

We thank the reviewers for their valuable comments. We also thank Haeyoon Cho, Jae-Won Chung, Jeongyoon Eo, Wonwook Song, Gyewon Lee, Soojeong Kim, Hokwen Joung, and Taegyun Kim for their constructive feedback. This work was partly supported by Google Research Award and Institute of Information & communications Technology Planning & Evaluation (IITP) grant funded by the Korea government (MSIT) [NO.2021-0-01343, Artificial Intelligence Graduate School Program (Seoul National University)].

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
