# Supplementary Material for Paper "Terra: Imperative-Symbolic Co-Execution of Imperative Deep Learning Programs"

## A    Criteria for Node Equality When Merging Traces

```
1  def function(x, N):
2    for i in range(N):
3      x = opA(x)
4    return x
```

```
1  def function_jit(x, N):
2    try:
3      terra_runtime_info.push_call_id(call_id)
4      try:
5        terra_runtime_info.push_loop_pair(
6            (loop_id, 0))
7        for i in range(N):
8          terra_runtime_info.inc_loop_counter(
9              loop_id)
10         x = opA(x)  # another func call
11     finally:
12       terra_runtime_info.pop_loop_pair()
13     return x
14   finally:
15     terra_runtime_info.pop_call_id()
```

(a) Original Program                    (b) Transformed Program

Figure 1: Conceptual illustration of how Terra applies JIT compilation to track a *call id* and a *loop id*

When the GraphGenerator attempts to match two operations while merging multiple traces into a TraceGraph, it compares the type, attributes, and the executed location of each operation. A type of an operation is a kind of the operation, and attributes of operations are information that determines the behavior of the operation. For example, the *MatMul* operation of TensorFlow has '*MatMul*' as its type, and takes *transpose_a* and *transpose_b* as the operation attributes to determine whether the input matrices should be transposed or not. If the GraphGenerator attempts to match the *MatMul* operation whose *transpose_a* is *true* with the *MatMul* operation whose *transpose_a* is *false*, the GraphGenerator fails to match because of the different attributes.

Each executed location of operations stands for the program location in which the operation is actually executed. Since the executed location of the operation is determined at runtime, Terra utilizes a just-in-time (JIT) compilation to evaluate the location. As shown in Figure 1, Terra assigns unique *call id*s to every function call and unique *loop id*s to every loop in a given imperative DL program. For each function call, the *call id* of the function is pushed to the *call id stack*, which accumulates the *call id*s. Terra manages the *call id stack* for the entire program execution [1], including the tracing phase and the co-execution phase. The pushed *call id* is popped when the function is returned. Thus, the *call id stack* contains all information of nested function calls. Similarly, the pair of (*loop id, loop counter=0*) of the loop is pushed to the *loop id stack* for each loop. The *loop counter* is increased for every new iteration of the loop, and the pair of (*loop id, loop counter*) is popped after exiting the loop. As same as the *call id stack*, Terra manages the *loop id stack* for the entire program execution.

---

[1]Current implementation of Terra does not consider multi-threading yet.

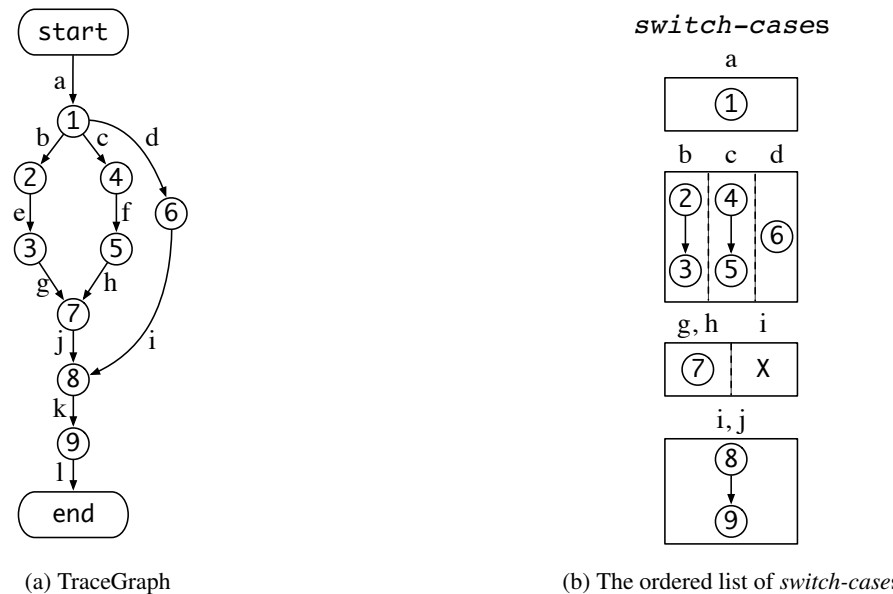

| (a) TraceGraph | (b) The ordered list of *switch-case*s |

Figure 2: The result of the case assignment algorithm for the given TraceGraph.

## B  Case Assignment Algorithm

In this section, we describe the case assignment algorithm that Terra uses to explicitly insert the *Switch-Case* operations in the symbolic graph. The algorithm takes a TraceGraph as an input and returns an ordered list of *switch-case*s. A *switch-case* is a set of (*basic block*, *control edges*) where the *basic block* is a linear chain of nodes, and the *control edges* are the edges that point to the *basic block*. Every non-overlapping linear chain of nodes in the TraceGraph is uniquely assigned to a *basic block* so that the ordered list of *switch-case*s can cover every trace in the TraceGraph. If there is a loop node in the TraceGraph, the algorithm treats it as a single node because the loop node is converted to the *While* operation in the symbolic graph. For example, from the TraceGraph of Figure 2a, the algorithm returns the ordered list of *switch-case*s of Figure 2b.

Algorithm 1 describes how the case assignment algorithm works. The algorithm traverses the given TraceGraph in topological order and makes each *basic block* contain a linear chain of nodes as long as possible. Figure 3 shows an example workflow of the algorithm when the TraceGraph of Figure 2a is an input. At first, the *next_edges* is initialized with {*edge a*} at line 2. Then the algorithm calculates the in-degree of *node 1* from $E \setminus next\_edges$ at line 12. Since *node 1* has no more incoming edge except for *edge a*, it becomes the first node of *basic_block* at line 14. Then the algorithm attempts to expand *basic_block* as long as possible, but it cannot expand because the out-degree of *node 1* is 3 so that *node 1* is the end of the linear chain (line 16). Thus, the first *switch-case* becomes ({*node 1*}, {*edge a*}) at line 25. At the next iteration, the *next_edges* becomes {*edge b*, *edge c*, *edge d*}, and three *basic block*s are created in the single *switch-case*. Two of them contain the linear chain with two nodes–{*node 2*, *node 3*} and {*node 4*, *node 5*}–and the last *basic block* contains {*node 6*}. When the algorithm processes *edge i* along with {edge g, edge h}, it does not put *node 8* into the *basic block* because the in-degree of *node 8* is not zero (line 12) due to *edge j*. Thus, the *basic block* becomes an empty set. Finally, the algorithm returns the ordered list of *switch-case*s after creating the *basic block* with *node 8* and *node 9*.

As shown in Figure 3, each *switch-case* within the result of the case assignment algorithm becomes the *Switch-Case* operation in the symbolic graph. If a *switch-case* contains only a single *basic block*, the GraphGenerator does not create a redundant *Switch-Case* operation. For each *Switch-Case* operation, the GraphGenerator creates the *Case Select* operation. During the co-execution, the PythonRunner informs the GraphRunner of the control edge taken via the *Case Select* operation. For example, if the PythonRunner follows *edge c* of Figure 3, the GraphRunner executes *case 2* of the first *Switch-Case* operation.

**Algorithm 1:** Terra's case assignment algorithm.

---

**Input:** TraceGraph $G = (V \cup \{start, end\}, E)$ where $V = \{v_1, v_2, \ldots, v_n\}$ and $E = \{e_1, e_2, \ldots, e_m\}$
**Output:** An ordered list of *switch-case*s $S = [s_1, s_2, \ldots, s_p]$

1  $S \leftarrow []$
2  next_edges $\leftarrow \{ (start, y) \in E \mid y \in V \}$
3  *// create a switch_case for each iteration*
4  **while** next_edges $\neq \emptyset$ **do**
5      switch_case $\leftarrow \emptyset$
6      new_next_edges $\leftarrow \emptyset$
7      *// create a case of switch_case for each $v$*
8      **forall** $v \in \{ y \mid (x, y) \in$ next_edges$\}$ **do**
9          control_edges $\leftarrow$ edges that point to $v$ among next_edges
10         basic_block $\leftarrow \emptyset$
11         *// no incoming edges to $v$ from $E \setminus$ next_edges*
12         **if** in-degree$(v, E \setminus$ next_edges$) = 0$ **then**
13            *// add $v$ to the basic block*
14            basic_block $\leftarrow$ basic_block $\cup \{v\}$
15            *// expand basic block to contain the linear chain as long as possible*
16            **while** out-degree$(v) = 1$ and in-degree$($next$(v)) = 1$ and next$(v) \neq end$ **do**
17               $v \leftarrow$ next$(v)$
18               basic_block $\leftarrow$ basic_block $\cup \{v\}$
19            *// collect new edges from $v$*
20            new_next_edges $\leftarrow$ new_next_edges $\cup \{ (v, y) \in E \mid y \in V \}$
21         **else**
22            *// keep control_edges for the next iteration of the outer while loop*
23            new_next_edges $\leftarrow$ new_next_edges $\cup$ control_edges
24         *// update switch_case*
25         switch_case $\leftarrow$ switch_case $\cup \{($basic_block, control_edges$)\}$
26      $S$.ListAppend(switch_case)
27      next_edges $\leftarrow$ new_next_edges
28  **return** $S$

---

Now we describe the formal definitions and the proof of the correctness of the algorithm.

**Definition 1.** A **TraceGraph** $G = (V \cup \{start, end\}, E)$ is a directed acyclic graph (DAG) where $V$ is set of nodes ($V = \{v_1, \ldots, v_n\}$) and $E$ is set of directed edges ($E = \{e_1, \ldots, e_m\}$) that connect the nodes. The TraceGraph has two extra nodes: the *start* node and the *end* node. The *start* node is a unique source node (i.e., in-degree of the *start* node is 0) and the *end* node is a unique sink node (i.e., out-degree of the *end* node is 0) of the TraceGraph.

**Definition 2.** A **linear chain** is an ordered set of nodes $L = \{v_1, \ldots, v_l\} \subseteq V$ such that for all $2 \leq i \leq l$, $(v_{i-1}, v_i) \in E$, the in-degrees of all nodes are 1 except $v_1$, and the out-degrees of all nodes are 1 except $v_l$. Also, the ordered set of edges in the linear chain, $I(L) = \{ (v_{i-1}, v_i) \mid 2 \leq i \leq l \}$, is called **in-chain edges**.

**Definition 3.** A **case** $c$ is a pair of (**basic block**, **control edges**) = $(L_c, E_c)$ where $L_c = \{v_1, \ldots, v_l\}$ is a *linear chain* and $E_c$ is a subset of $E$ with the edges that point to $v_1$ of $L_c$. In addition, a **switch-case** $s$ is a set of *case*s that satisfies the following condition:

$$\forall c_1, c_2 \in s \text{ such that } c_1 \neq c_2, L_{c_1} \cap L_{c_2} = \emptyset \text{ and } E_{c_1} \cap E_{c_2} = \emptyset.$$

In other words, different *case*s are mutually exclusive.

**Definition 4.** A **trace** $t = (V_t \cup \{start, end\}, E_t)$ is a DAG that satisfies the following conditions:

1. $V_t = \{v_1, \ldots, v_{k-1}\} \subseteq V$ and $E_t = \{e_1, \ldots, e_k\} \subseteq E$
2. $\forall 1 \leq i \leq k, e_i = (v_{i-1}, v_i)$ where $v_0 = start, v_k = end$

Moreover, the **operation nodes** $Op(t)$ of the *trace* $t$ is $V_t$, and the **path** $Path(t)$ of the *trace* $t$ is $E_t$.

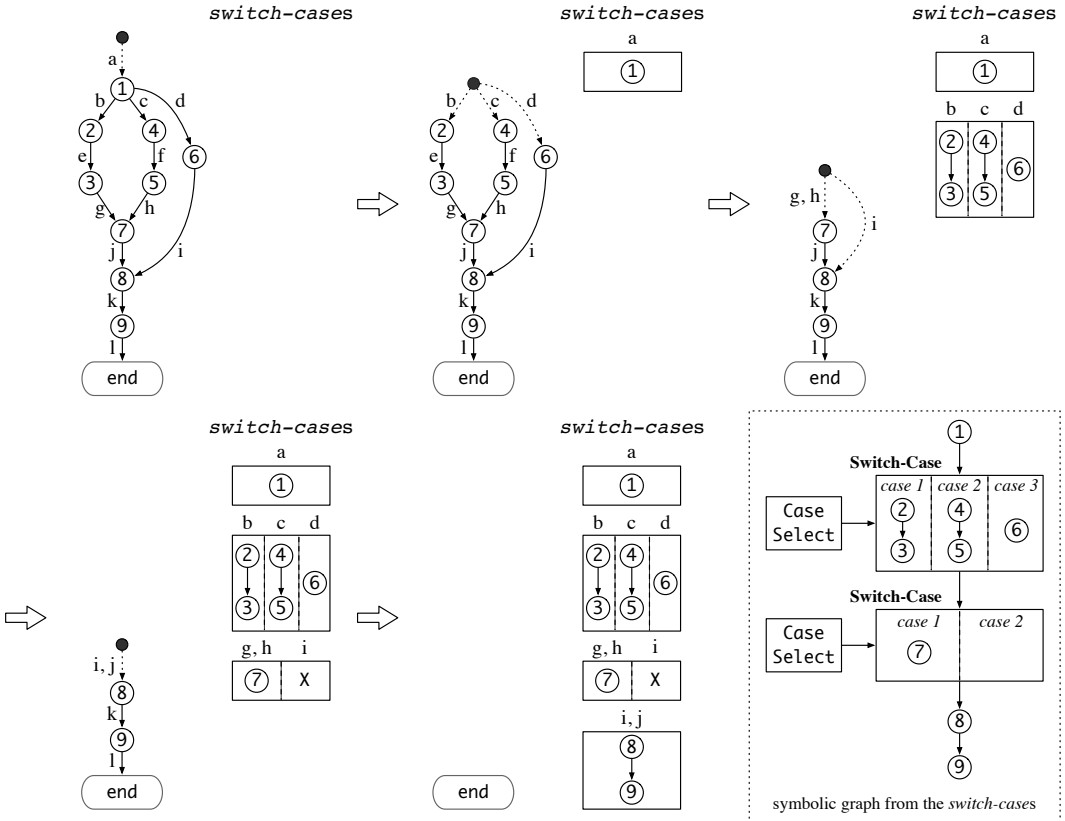

Figure 3: Example workflow of how the case assignment algorithm works, and how the symbolic graph is generated from the ordered list of *switch-case*s. The dotted arrows from a black circle denote the *next_edges* variable of Algorithm 1. All processed nodes and edges are assigned to an appropriate *switch-case*, where each rectangle of the *switch-case*s denotes the *basic block*. X denotes that no node exists in the *basic block*.

**Definition 5.** An ordered list $S = [s_1, \ldots, s_p]$ of *switch-case*s covers a *trace* $t$ if the following conditions hold.

1. For all $s_i$, there exists a unique *case* $c_i = (L_{c_i}, E_{c_i}) \in s_i$ with a unique edge $d_i$ such that $\{d_i\} = E_{c_i} \cap Path(t)$. The other *case*s do not have corresponding *control edges* for $Path(t)$.

2. The operations in the trace are represented as the linear chains of the $c_i$'s, and all edges in the trace are the union of *in-chain edges*, the $d_i$'s, and $e_k$. That is,

$$Op(t) = \bigcup_{i=1}^{p} L_{c_i} \text{ and } Path(t) = \left[ \bigcup_{i=1}^{p} (\{d_i\} \cup I(L_{c_i})) \right] \cup \{e_k\}.$$

Note that $e_k$ is the edge that points to the *end* node.

**Definition 6.** A graph $G_s = (V_s \cup \{start, end\}, E_s)$ is a **sub-TraceGraph** with $V_s$ if

1. $G_s$ is a TraceGraph and $V_s \subseteq V$.

2. $E_s = E_1 \cup E_2$ where

$$E_1 = \{ (u, v) \in E \mid u \in V_s \cup \{start\}, \ v \in V_s \cup \{end\} \}$$
$$E_2 = \{ (u, end) \mid (u, v) \in E, \ u \in V_s \cup \{start\}, \ v \notin V_s \cup \{end\} \}.$$

To be specific, $E_1$ denotes all the edges between the nodes within $V_s \cup \{start, end\}$. Furthermore, for all edges whose source node $u$ is in $V_s \cup \{start\}$ and destination node $v$ is not in $V_s \cup \{end\}$,

the sub-TraceGraph changes the destination node of such edges to the *end* node because the *end* node should be a unique sink node of the TraceGraph. Then, $E_2$ denotes the changed edges. We define the sub-TraceGraph to use in the proof of the following theorem.

**Theorem 1.** Algorithm 1 generates an ordered list $S$ of *switch-case*s that covers every *trace* in the TraceGraph $G = (V \cup \{start, end\}, E)$.

***Proof of Theorem 1.*** To prove the theorem, we define the following auxiliary variables:

- *processed nodes* $V_p = \bigcup_{s \in S} \bigcup_{c \in s} \{\, v \in L_c \mid c = (L_c, E_c) \,\}$
- *connecting edges* $C = \{\, (u, v) \in E \mid u \in V_p \cup \{start\}\,, v \notin V_p \cup \{end\}\}$
- *processed TraceGraph* $G_p = (\, V_p \cup \{\, start, end\,\}, E_p \,)$

First, the *processed nodes* is a set of all nodes in $S$, which is the same as a set of the nodes that the algorithm visited. The *connecting edges* is a set of edges where the source node of each edge is in $V_p \cup \{start\}$ and the destination node of each edge is not in $V_p \cup \{end\}$. Finally, the *processed TraceGraph* represents the TraceGraph with the *processed nodes*. It has $E_p$ such that

$$E_p = \{\, (u, v) \in E \mid u \in V_p \cup \{start\}, \ v \in V_p \cup \{end\}\,\} \cup$$
$$\{(u, end) \mid (u, v) \in C\}.$$

Then, we use the following loop invariants prior to every iteration of the loop at line 4.

1. The *next_edges* variable of the algorithm is identical to the *connecting edges* $C$.
2. The *processed TraceGraph* $G_p$ is a *sub-TraceGraph* with $V_p$.
3. The variable $S$ is an ordered list of *switch-case*s, and it covers every *trace* in $G_p$.

At the beginning of the loop, the three loop invariants hold with $S = []$, $V_p = \emptyset$, $C = \{\, (u, v) \in E \mid u = start, v \neq end\,\}$, and $E_p = \{(start, end)\}$.

Next, we prove that the loop invariants are maintained after each iteration. For each $v$ at line 8, the variable *basic_block* is a *linear chain* collected throughout the while loop of line 16 or an empty set. If the *basic_block* contains a *linear chain*, $V_p$ adds all the nodes within the *linear chain*. Then the *next_edges* becomes the new edges that point to the nodes which are not in $V_p \cup \{end\}$ (line 20). If the *basic_block* is an empty set, it denotes that $v$ is not added to $V_p$. Then, the *next_edges* becomes the *control_edges*, where all the edges are pointing to $v$ (line 23). Thus, the first loop invariant holds. Moreover, the second loop invariant holds because the *linear chain*s of each iteration extend $V_p$ with corresponding nodes while including *in-chain edges* and updating the *connecting edges*. Subsequently, the third loop invariant holds because each *control edges* is assigned to the specific *case* with the corresponding *linear chain* (line 25).

Finally, we prove the theorem by showing the following propositions are true.

1. For each iteration, $|V_p|$ strictly increases.
2. After the termination of the outermost while loop, $V_p = V$ and $G_p = G$.

The first proposition shows that the outermost loop eventually finishes, and the second proposition shows that the variable $S$ covers every *trace* in $G$. First of all, for an iteration, let $N = \{v_1, v_2, \ldots, v_{|N|}\}$ from $\{\, y \mid (x, y) \in \text{next\_edges}\}$ at line 8. Then, suppose that $|V_p|$ is not increased, which denotes

$$\forall v_i \in N, \text{in-degree}(v_i, E \setminus \text{next\_edges}) \neq 0$$

at line 12. In other words, it implies that

$$\forall\, v_i, \exists\, v_j \in N \text{ such that } v_j \neq v_i, v_j \sim v_i$$

where $x \sim y$ indicates that for $x \in V$ and $y \in V$, there exists a path from $x$ to $y$ in $G$. Without loss of generality, assume that $v_2 \sim v_1$. Then, there should exist $j$ such that $3 \leq j \leq |N|$ and $v_j \sim v_2$. However, this requires a cycle in $G$ in the end, which contradicts to the assumption: $G$ is a DAG. Thus, $|V_p|$ strictly increases throughout the iterations.

```
1  import tensorflow as tf
2
3  for inputs, labels in train_data_loader:
4    with tf.TerraGradientTape() as tape:
5      logits = model(inputs)
6      loss = loss_fn(logits, labels)
7    grads = tape.gradient(
8        loss, model.trainable_variables)
9    optimizer.apply_gradients(
10       zip(grads, model.trainable_variables))
```

Figure 4: Programming interface of Terra

Now suppose that $V_p \nsubseteq V$ after the termination of the outermost loop. Then, there exists $v \in V \setminus V_p$. However, this contradicts to the termination condition of the loop because there exists a path from *start* to $v$ by the definition of the TraceGraph. Thus, $V_p = V$ and $G_p = G$ after the termination of the outermost while loop. $\square$

## C  Implementation Details

### C.1  Modification to TensorFlow's Internal System

We modified the imperative execution model of TensorFlow for both GraphGenerator and Python-Runner. When the Python interpreter executes DL operations imperatively in the tracing phase, the interpreter makes the GraphGenerator record each operation as a symbolic representation, which is a *NodeDef* of TensorFlow. During the co-execution phase, functions that trigger an actual computation of a DL operation (`TFE_Py_Execute` function and `TFE_Py_FastPathExecute_C` function) are modified to perform validating the symbolic graph and creating an empty tensor object. To annotate feed points, we modified `FuncGraph.capture` to capture all external tensors. Similarly, we modified `EagerTensor.numpy` to annotate fetch points.

### C.2  Programming Interface

Figure 4 shows an example code of using Terra to speed up the training process. All the programmers have to do is just to modify a single line of code in their imperative DL program: changing from `tf.GradientTape` to `tf.TerraGradientTape` at line 4. Since all imperative TensorFlow programs must use `GradientTape` to train DNNs, Terra is applicable to all imperative programs transparently without programmers' extra burden. Terra generates the symbolic graph from the DL operations within the `TerraGradientTape` context and the gradient computations (`tape.gradient`).

### C.3  Communication Between the PythonRunner and the GraphRunner

Although the PythonRunner executes the skeleton imperative program sequentially, the graph executor of the GraphRunner allows out-of-order execution. Thus, a deadlock could occur if the two Runners conduct the co-execution naively. Suppose that Terra executes the imperative program shown in Figure 5a. Terra generates the symbolic graph from the imperative program as shown in Figure 5b. Since the two operations do not have data and control dependency in the symbolic graph (i.e., *opB* does not consume *opA*'s output), the GraphRunner can freely select the execution order between the operations. If the GraphRunner executes *opB* then *opA*, the deadlock would occur because the PythonRunner should receive the output of *opA* to print its value before it feeds the value *k* to *opB* in the GraphRunner. To prevent the deadlock, the GraphGenerator adds the control dependencies (defined in TensorFlow) between *Output Fetching* operations that should be executed prior to and an *Input Feeding* operation after generating a symbolic graph. Since the TraceGraph of Terra captures the execution orders between the collected operations, the GraphGenerator can figure out the control dependencies.

```
1   x = opA()
2   print(x)
3   k = tf.constant(random.random())
4   # k becomes InputFeeding in the sym. graph
5   y = opB(k)
```

(a) Imperative Program

(b) Symbolic Graph

Figure 5: Possible case of deadlock if Terra does not add control dependency between the *Output Fetching* and the *Input Feeding* operations. Note that there is no data dependency between *opA* and *opB* in the symbolic graph.

## C.4  Fallback Handling

When the PythonRunner detects a new trace in the co-execution phase, Terra cancels the execution of the GraphRunner. Then, the PythonRunner executes all the DL operations that have been matched within the current iteration to make the program state as if it were being performed imperatively from the beginning. While executing the matched operations, some of them could be executed twice if the GraphRunner already executed the operations. This can be a problem for stateful operations, which hold and change the program state such as I/O operations and communication operations. To prevent this problem, stateful operations are not recorded by the GraphGenerator so that those operations are not included in the symbolic graph. Any inputs and outputs of the stateful operations are connected with the symbolic graph through the *Input Feeding* and the *Output Fetching* operations.

We exceptionally allow the GraphGenerator to record and generate stateful operations that are related to variables (both trainable and non-trainable) of a DNN such as the *ReadVariableOp* operation and the *AssignVariableOp* operation of TensorFlow to optimize performance. To ensure correct execution, the GraphGenerator inserts control dependencies between those operations (e.g., ensuring read after write) automatically while generating the symbolic graph. Furthermore, for the variables whose update operations are generated in the symbolic graph (e.g., updating moving averages of batch normalization), the PythonRunner makes a checkpoint of those variables at the beginning of each iteration. Whenever the GraphRunner's execution is canceled, the PythonRunner restores such variables from the checkpoint and executes the operations that the PythonRunner has succeeded to match.

## D  Details on the AutoGraph Failure Cases

Figure 6 shows the codes that AutoGraph fails to convert. First, DropBlock [4] keeps `keep_prob` in the class object and alters it during training. However, AutoGraph cannot detect the mutation throughout the training. Similarly, AutoGraph cannot capture the object mutations of both Music-Transformer [8] and SDPoint [9]. For MusicTransformer, the object mutation is not related to the algorithmic characteristic of the model but the programming style of the user. It wraps the entire training process in a single *trainer* class, which is a common design pattern for implementing a program that trains a DNN [5]. The `_train_step` method calculates the loss value of the model for each training step, and it writes the value to the `loss_value` attribute (line 6 of Figure 6b). However, AutoGraph cannot write the new loss value to the attribute because it does not access the Python heap while carrying out the symbolic execution. Thus, when the Python interpreter attempts to read `loss_value` (line 12), it fails to read the updated loss. Terra correctly captures those mutations because the PythonRunner accesses the Python heap and updates the objects in the *co-execution* phase. BERT-CLS [10] and FasterRCNN [19] show the example cases of *tensor materialization during conversion*. For both cases, AutoGraph fails to generate a graph because they cannot evaluate the exact value of the tensors while generating the symbolic graph. Moreover, BERT-CLS should evaluate the tensor values to calculate the target metric via a third-party library [3], which Auto-Graph does not support. However, Terra is not affected by such cases because the GraphGenerator collects traces while Terra carrying out the imperative execution in the tracing phase. Then in the co-execution phase, the PythonRunner materializes those tensors via the *Output Fetching* operations of the symbolic graph.

```python
1  class DropBlock2D(tf.keras.layers.Layer):
2    def set_keep_prob(self, keep_prob=None):
3      if keep_prob is not None:
4        self.keep_prob = keep_prob
5      w, h = tf.cast(self.w, tf.float32), \
6          tf.cast(self.h, tf.float32)
7      self.gamma = (1. - self.keep_prob) * (w * h) \
8          ...
9   def call(self, x, training=False):
10     ...
11     mask = _bernoulli(sampling_mask_shape, self.gamma)
12     ...
```

(a) DropBlock

```python
1  class MusicTransformerDecoder(tf.keras.models.Model):
2    def _train_step(self, inp_tar, out_tar,
3                    lookup_mask, training):
4      predictions = self.call(inp_tar, lookup_mask,
5                              training)
6      self.loss_value = self.loss(out_tar, predictions)
7      ...
8
9    def train_on_batch(self, ...):
10     # not a conversion scope
11     ...
12     loss = tf.reduce_mean(self.loss_value)
```

(b) MusicTransformer

```python
1  class SDResNet(tf.keras.models.Model):
2    def stochastic_downsampling(self, blockID, ratio):
3      downsampling_ratio = ratio is None and 0.5 or ratio
4      for l in self.layers:
5        if isinstance(l, _ConvBlock):
6          if l.blockID == blockID:
7            l.downsampling_ratio = downsampling_ratio
8          else:
9            l.downsampling_ratio = 1.
```

(c) SDPoint

```python
1  from sklearn.metrics import f1_score
2  class SparseF1Score(object):
3    def __call__(self, y_true, y_predict):
4      y_true = tf.reshape(
5          tf.constant(y_true), [-1]).numpy()
6      y_predict = tf.reshape(
7          tf.argmax(y_predict, -1), [-1]).numpy()
8      f1 = f1_score(y_true, y_predict,
9          average=self.average)
10     return f1
```

(d) BERT-CLS

```python
1  def calc_batch_padded_shape(meta):
2    return tf.cast(tf.reduce_max(
3        meta[:, 6:8], axis=0), tf.int32).numpy()
```

(e) FasterRCNN

Figure 6: Code snippets that AutoGraph fails to convert correctly.

Table 1: Details on training throughput evaluation.

| Program | Dataset | Batch Size | Throughput (per second) |
|---------|---------|------------|-------------------------|
| ResNet50 [17] | ImageNet [14] | 64 | 188.02 ($\pm$ 1.08) images |
| BERT-Q&A [7] | SQuAD1.1 [13] | 4$\times$384 | 6184.49 ($\pm$ 20.24) tokens |
| YOLOv3 [20] | VOC2012 [6] | 8 | 30.00 ($\pm$ 0.37) images |
| DCGAN [16] | MNIST [12] | 500 | 10727.19 ($\pm$ 22.25) images |
| GPT2 [15] | OpenWebText [2] | 8$\times$515 | 8764.26 ($\pm$ 94.20) tokens |
| BERT-CLS [10] | NLPGNNDATA [11] | 512$\times$4 | 9443.13 ($\pm$ 34.03) tokens |
| DropBlock [4] | ImageNet | 64 | 185.23 ($\pm$ 0.28) images |
| SDPoint [9] | ImageNet | 32 | 94.33 ($\pm$ 1.59) images |
| FasterRCNN [19] | COCO2017 [18] | 1 | 2.56 ($\pm$ 0.03) images |
| MusicTransformer [8] | e-Piano MIDI [1] | 2$\times$2048 | 20031.08 ($\pm$ 48.49) samples |

Table 2: Results of the number of collected traces and the number of fallbacks for each program.

| Program | # Collected Traces | # Fallbacks |
|---------|--------------------|-------------|
| ResNet50 [17] | 2 | 0 |
| BERT-Q&A [7] | 2 | 0 |
| YOLOv3 [20] | 2 | 0 |
| DCGAN [16] | 2 | 0 |
| GPT2 [15] | 2 | 0 |
| BERT-CLS [10] | 2 | 0 |
| DropBlock [4] | 3 | 0 |
| SDPoint [9] | 4 | 1 |
| FasterRCNN [19] | 2 | 0 |
| MusicTransformer [8] | 2 | 0 |

# E   Details on Training Throughput Evaluation

Table 1 shows the absolute throughput values of Terra for the evaluated programs. Batch sizes of BERT-Q&A, GPT2, BERT-CLS, and MusicTransformer denote batch_size $\times$ sequence_length, which is the number of tokens in a batch of transformer-based models. DropBlock and SDPoint use ImageNet because they utilize ResNet50 as their backbone network.

# F   Tracing Phase Analysis

Table 2 shows the number of collected traces and the number of fallbacks from the *co-execution* of Terra. The results show that the symbolic graphs of all the programs can be generated with at most four traces. This indicates that our approach–collecting multiple traces and incrementally generating the symbolic graph–is a plausible strategy. Furthermore, the number of collected traces for BERT-CLS, FasterRCNN, and MusicTransformer shows that Terra correctly executes the programs with only two traces, while AutoGraph cannot execute them at all.