# OpenReview forum: "Terra: Imperative-Symbolic Co-Execution of Imperative Deep Learning Programs"
_NeurIPS.cc/2021/Conference — NeurIPS 2021 Poster_

### Official Review · Reviewer_FiWe · 2021-07-13

**Rating:** 6
**Confidence:** 3

**Summary:**

This work proposes an approach for achieving optimized performance of symbolic graph execution for any imperative DL program. The approach is implemented in a system called Terra. The proposed approach involves decoupling symbolic operations and building a symbolic graph. The skeleton python program is executed using the Python interpreter whereas the symbolic operations are delegated to the optimized symbolic execution. Terra is evaluated on ten imperative DL programs and the performance is compared with AutoGraph. Terra can successfully execute all the ten programs whereas AutoGraph fails to execute five of them. The speed up achieved by Terra is comparable to AutoGraph. The main appeal of the method is that it does not have any Python feature coverage limitations.


**Limitations And Societal Impact:**

I believe that more discussion is needed on limitations or possible pain points of the proposed approach. For example, one can ask:
- Does the method put any limitations on the kind of graphs that can be generated? Does it miss out on certain optimization opportunities due to the nature of the graph generation.
- Are there use cases where AutoGraph will perform significantly better than Terra?
- For the programs where AutoGraph works, does it make sense to use Terra considering that the performance on Terra would be slightly worse than AutoGraph?
- Does the way the program is implemented have any impact on the performance of Terra? (Possibly due to excessive switching between the two running modes.)

The answers to these questions will either bolster the claims or warn the readers about possible limitations.

I do not believe that there is any potential negative societal impact of the work.

**Main Review:**

### Originality:
The work tries to combine the "imperative execution" and "symbolic execution" paradigms to get the best of both worlds. Instead of trying to convert the whole program to a symbolic graph, it just decouples the symbolic operations into a symbolic graph. The control flow is governed by the skeleton Python program executed in an imperative way whereas the symbolic graph is executed by a Symbolic Runner. Although the idea of speculative generation of symbolic graphs has been explored in earlier works, the idea of co-execution is novel. Co-execution is crucial for ensuring support for all the host language features.

### Quality:

The paper is technically sound.

* The work has done a good job of carefully designing the constructs for communicating data and managing control flow across the two runners.
* The proof that the graph generation algorithm can handle any DAG is described in the supplementary material.
* To support the claim that Terra performs comparably with Autograph on the programs on which AutoGraph works, it would be preferable to see the numbers on the benchmarks from the AutoGraph paper. The rationale for selecting the 10 programs is not discussed. It would be nice if the authors could discuss different dimensions they tried to cover( architectures, type of input data, etc.)
* The paper compares Terra with a single competing method(AutoGraph). I would have liked to see comparisons with other methods from the "single path tracing" and "static compilation paradigms".


### Clarity
The paper is well written and easier to follow along.

Some minor issues.
* I do not understand when "GraphGenerator" stops collecting the traces (line 190). A little elaboration would be helpful.
* When Terra cancels the execution of the GraphRunner and falls back to the tracing phase (line 212), is the computation done so far lost?
* In section 4.2 (TraceGraph), in almost all the sentences, the word "GraphGenerator" is repeated. "GraphGenerator attempts ...", "GraphGenerator compares...", etc. Since the whole section is about GraphGenerator, I found the repeated usage a bit distracting. It is a style issue; authors may want to improve it.
* Spelling mistake. Line 307. "publicy" and "opended"
* Line 190, "lastest"

### Significance

The paper is trying to address a very important problem. The main appeal of the "define-by-run" imperative programs is the flexibility. The prevalent approaches which try to optimize the performance of imperative programs end up putting restrictions on the program features that can be used. Terra's approach of co-execution allows it to support all the host language features without sacrificing the performance. This paper will inspire more work in this direction.

Although this is not discussed in the paper, I believe that since the control flow is governed by the python interpreter, this approach will have better debuggability than the approaches that try to convert the whole program to a symbolic graph.


**Time Spent Reviewing:**

8

---

> ### Author Response · Authors · 2021-08-10
> **Response to Reviewer FiWe**
>
> We thank the reviewer for the comments.
>
> ### Quality
>
> To choose programs for the evaluation, we considered the programs that are implemented based on TensorFlow's imperative API.
> Among such programs, we selected representative implementations of various tasks such as image classification, object detection, image generation, sound processing, and natural language processing to compare the performance of imperative execution, AutoGraph, and Terra.
>
> Comparing with the single path tracing is possible by disabling AutoGraph's static compilation (`tf.function(autograph=False)`).
> For the five programs that AutoGraph can handle, the performance of the single path tracing is same as that of AutoGraph.
> For the five programs that AutoGraph cannot handle, the single path tracing cannot also handle them.
> To compare with the static compilation approach, we might use TorchScript, the only framework that is publicly available among the static compilation approaches. However, since TorchScript supports only PyTorch, it is hard to conduct an apple-to-apple comparison because of the framework difference.
>
>
> ### Clarity
>
> We would like to clarify when the `GraphGenerator` stops collecting the traces.
> The `GraphGenerator` checks whether a new trace is covered by the current TraceGraph or not.
> In other words, if the new trace is a subgraph of the TraceGraph, the `GraphGenerator` determines that it has collected enough traces to generate a symbolic graph.
> While checking that the new trace is a subgraph of the TraceGraph or not, Terra compares the type of operation, attributes of operation, and whether two operations were executed at the same location of the program as described in the paper.
> Furthermore, if two operations satisfy the equality criteria but the shapes are different, the `GraphGenerator` does not treat them as different operations.
> Instead, it uses the dynamic shape representation of TensorFlow (i.e., `None` shape).
> Therefore, the `GraphGenerator` can build a symbolic graph with a small number of traces.
>
> The current implementation of Terra discards the computation done so far when it cancels the execution of the `GraphRunner`.
> We adopt this policy to make use of the automatic differentiation of TensorFlow's imperative execution [1] transparently.
>
> [1] Akshay Agrawal, Akshay Naresh Modi, Alexandre Passos, Allen Lavoie, Ashish Agarwal, Asim Shankar, Igor Ganichev, Josh Levenberg, Mingsheng Hong, Rajat Monga, and Shanqing Cai. TensorFlow Eager: A Multi-Stage, Python-Embedded DSL for Machine Learning. In Proceedings of Conference on Systems and Machine Learning (SysML), 2019.
>
> ### Limitations
>
> > Does the method put any limitations on the kind of graphs that can be generated? Does it miss out on certain optimization opportunities due to the nature of the graph generation.
>
> We extended TensorFlow's graph compiler (i.e., Grappler) to handle `InputFeeding` and `OutputFetching`, so Terra does not miss graph optimization opportunities when it executes a symbolic graph with the TensorFlow backend.
> When applying XLA to Terra, there is a possibility to miss some optimization opportunities because the current XLA implementation does not convert the custom operations of Terra (i.e., `InputFeeding` and `OutputFetching`) to HLO IR.
> However, we can extend XLA implementation to convert our custom operations to HLO IR to fully support XLA optimization.
>
> > Are there use cases where AutoGraph will perform significantly better than Terra?
>
> AutoGraph can perform better than Terra when the Python Interpreter has much code to interpret while the underlying DL model is not compute-intensive.
> In that case, Terra's execution time is bounded by the PythonRunner.
> Nevertheless, Terra still performs better than imperative execution even in such cases.
>
> > For the programs where AutoGraph works, does it make sense to use Terra considering that the performance on Terra would be slightly worse than AutoGraph?
>
> If AutoGraph works correctly, we think that programmers can use AutoGraph as well.
> However, if programmers are not certain of the correctness of the AutoGraph conversion or they want to avoid some extra works for using AutoGraph (e.g., writing an input signature and refactoring code by considering conversion scopes of AutoGraph), Terra is a better choice than AutoGraph.
>
> > Does the way the program is implemented have any impact on the performance of Terra? (Possibly due to excessive switching between the two running modes.)
>
> We agree that Terra's performance can degrade when the imperative DL program incurs excessive switching between the two running modes.
> We have not observed such programs with excessive switching yet.
> For such programs, AutoGraph is likely to fail to compile because the existence of excessive switching in Terra implies that the program uses excessive Python features that AutoGraph cannot deal with.

---

### Official Review · Reviewer_4LTS · 2021-07-16

**Rating:** 7
**Confidence:** 5

**Summary:**

This paper proposes a method for obtaining a symbolic (graph) representation of a model, given the imperative Python code. It distinguishes itself from (most) other methods by co-executing the symbolic as well as the imperative representation, as well as merging multiple traces into a single symbolic representation. This allows it to obtain arbitrary representations of



**Limitations And Societal Impact:**

Yes

**Main Review:**

Overall, I think this paper tackles a very relevant problem, and some parts of the approach are novel (to my knowledge) and quite interesting. Before I talk about this paper's strengths, I want to note some very related work that this paper should definitely have compared to. Specifically, that it is the LazyTensor approach (https://arxiv.org/abs/2102.13267), which does something similar, in that it builds up a symbolic graph while executing, and will recompile if that symbolic graph doesn't match. In this way, LazyTensor is also able to guarantee that it executes all code correctly.

Back to this paper, I am fairly impressed with the built system as well as the approach. AFAIK, I haven't seen this kind of "trace out program and merge traces into a single graph" approach before. Their results show that their approach is successful on a wide variety of models, including some models with a significant amount of dynamism.

My primary concern with this approach is quite complicated, and the co-execution of the symbolic and the imperative models seems to me to impair debuggability and UX. This isn't a fatal flaw by any means - if it can execute user code reliably enough then this isn't a problem - but is something to keep in mind with these kinds of approaches.

Another question I had was how the symbolic graph was actually being executed. In TF/PyTorch, one of the primary reasons to create a symbolic graph is to pass it to a compiler like XLA. However, it seems to me like this co-execution structure/python fallbacks in order to resolve control flow may 1. not allow you to use such compilers, 2. inhibit possible optimizations. Could you clarify whether this understanding is correct and whether the baseline AutoGraph was then lowered to XLA?

Overall, I find the approach interesting + think it tackles an important problem. However, I think lack of proper comparison to relevant work are somewhat problematic.


EDIT: Primarily based off of the new results presented with XLA, as well as the preliminary LazyTensor comparisons, I have raised my score from 5 to 7.

**Time Spent Reviewing:**

5

---

> ### Author Response · Authors · 2021-08-10
> **Response to Reviewer 4LTS**
>
> We thank the reviewer for the comments.
>
> We thank the reviewer for letting us know LazyTensor, which is an interesting related work. We think that LazyTensor and Terra are **concurrent** related work, and they adopt different approaches.
> In a nutshell, LazyTensor uses a lazy evaluation approach, in which the Python Interpreter and symbolic graph executor run alternatively.
> The Python Interpreter executes an imperative DL program as it is and extracts a linear trace of operations (an IR graph in the LazyTensor paper).
> LazyTensor then checks whether the extracted graph is already cached or not.
> If cached, LazyTensor directly executes the cached graph.
> If not, LazyTensor compiles and executes the new graph, then stores the graph for further iterations.
> However, since the Python Interpreter waits until the graph execution to be finished, it might entail an avoidable performance overhead.
> Furthermore, when LazyTensor encounters a tensor materialization point (e.g., a `print` function call, a tensor value that determines the control flow of the host language, and an explicit use of the `LazyTensorBarrier` annotation),
> LazyTensor may split an IR graph.
> Splitting a symbolic graph into multiple subgraphs can cause to miss some optimization opportunities.
> Finally, if a program has a dynamic control flow that is not determined by a tensor value, LazyTensor would fail to capture the control flow.
> The IR graph of each iteration could be different, but LazyTensor could not detect it without a programmer's `LazyTensorBarrier` annotation or replacing the control flow with a proper IR representation that could be captured.
>
> There are multiple differences between LazyTensor and Terra.
> Actually, our initial design was based on lazy evaluation, but we switched to co-execution to avoid the overhead.
> Instead of lazy evaluation, Terra co-executes the `PythonRunner` and the `GraphRunner`.
> During the co-execution, both `PythonRunner` and `GraphRunner` undertake their execution concurrently and one of the `Runner`s waits for the other only if they need the data from the other.
> Second, Terra does not split a symbolic graph at each tensor materialization point.
> Rather than splitting the graph, Terra keeps the graph as a single graph and fetches the value of a tensor through the `OutputFetching` operation. Therefore, Terra keeps the possibilities of optimizations intact.
> Lastly, since Terra merges multiple traces into a single TraceGraph and validates it for every iteration, Terra can deal with an arbitrary control flow without the extra works of programmers.
>
> In addition, the current version of the LazyTensor paper contains preliminary evaluation results.
> The work evaluates three models, RoBERTa, ResNet50, and WordSeg (Table 2, Table 3, and Table 4 respectively in the LazyTensor paper), and it compares LazyTensor only with PyTorch and the imperative execution of TensorFlow Swift. There is no comparison with static compilation or single path tracing approaches.
>
> For the concern about debuggability, Terra improves debuggability compared to AutoGraph.
> As Reviewer FiWe mentioned, since the Python Interpreter governs the program execution, Terra has better debuggability than the previous approaches that aim to convert the whole program to a symbolic graph.
> If an imperative DL program has an unhandled exception, both Terra and imperative execution raise the same exception with the same stack traceback.
> However, when the Python interpretation is omitted as in AutoGraph, it is hard to show the same stack traceback when an exception occurs as the imperative execution shows.
>
> For the final question, since Terra uses the GraphDef abstraction and graph executor of TensorFlow, Terra can make use of XLA as AutoGraph does.
> The following table shows the training speed up when applying XLA to both Terra and AutoGraph.
>
> | DL Program           | Terra               | Terra + XLA         | AutoGraph           | AutoGraph + XLA     |
> |:------------------|:-------------------:|:-------------------:|:-------------------:|:-------------------:|
> | ResNet50          | x1.25 | x1.38 | x1.25 | x1.40 |
> | BERT Q&A          | x1.20 | x1.42 | x1.26 | x1.48 |
> | YOLOv3            | x1.61 | x1.54 | x1.67 | x1.65 |
> | DCGAN             | x1.56 | x1.61 | x1.57 | x1.68 |
> | GPT2              | x1.27 | N/A   | x1.29 | N/A   |
> | BERT CLS          | x1.06 | x1.33 | N/A   | N/A   |
> | DropBlock         | x1.25 | x1.38 | N/A   | N/A   |
> | SDPoint           | x1.22 | x1.27 | N/A   | N/A   |
> | FasterRCNN        | x1.14 | N/A   | N/A   | N/A   |
> | Music Transformer | x1.30 | x1.73 | N/A   | N/A   |
>
> As reported in Figure 5 of the paper, each number denotes the amount of speed up relative to TensorFlow imperative execution.
> The values of 'Terra' and 'AutoGraph' are the same as the ones reported in Figure 5.
> We will update Figure 5 to include the results of Terra+XLA and AutoGraph+XLA to make clear that Terra can take advantage of additional XLA optimizations.
>
> When we apply XLA to Terra and AutoGraph, XLA is not applicable (N/A) to GPT2 and FasterRCNN due to the dynamic shape of the input data.
> For each training iteration, the shapes of input data to the models can change.
> However, XLA assumes static shapes, so it does not efficiently handle dynamic shapes; it compiles kernels every time the shape changes.
>
> As you can see, both Terra and AutoGraph achieve higher performance when applying XLA compilation except for YOLOv3.
> We profile the execution and find that the current XLA fails to efficiently cluster operations for YOLOv3, thus fails to create good kernels in YOLOv3.
> To be more specific, YOLOv3 includes some computation operations such as `ResizeNearestNeighbor` and `Where`, which are not supported by XLA. So, XLA cannot efficiently fuse computation operations.
> Additionally, we observe that Terra's performance decreases more than that of AutoGraph because the schedules of some `OutputFetching` operations are reordered because of XLA kernels, causing a longer stall in the `PythonRunner`.
> However, this problem can be addressed by extending XLA to support our `OutputFetching` operation.

---

> > ### Comment · Reviewer_4LTS · 2021-09-01
> > **Thanks for the comments**
> >
> > > We think that LazyTensor and Terra are concurrent related work
> > I'm not sure I agree with that, although I agree it's somewhat of a gray zone. Although the paper was put on Arxiv in February of this year, the underlying software (PyTorch/XLA) was released much earlier. A quick search shows that it was officially announced in September 2020.
> >
> > I'll leave it up to the AC to decide the appropriate course of action, but to me, a comparison to LazyTensor is still the primary missing piece of this paper. The comparisons to Autograph are neat, but the performance can mostly be summarized as "it performs the same, except when Autograph can't capture the graph".
> >
> > If I understand correctly, the practical advantage of Terra over LazyTensor is that there is less graph splitting/more capable of handling control flow. A comparison against LazyTensor would provide a program capture system that could actually be compared against Terra.
> >
> > Overall, however, I still find the ideas in the paper interesting. In particular, the notion of merging multiple traces together in order to capture the symbolic graph.
> >
> > Up to the AC, but if LazyTensor is removed from comparison, then I would rate this paper a 7. It would be the first system to combine "executing arbitrary Python code" with "lowering to a symbolic graph executor", which is a significant advantage beyond all current work. However, if LazyTensor is considered a related work, then the distinguishing factor of this paper is that it splits the subgraphs less than LazyTensor, and has some mechanisms to merge multiple traces into a single graph with control flow. And there, it suffers without a lack of comparison to LazyTensor.

---

> > > ### Author Response · Authors · 2021-09-03
> > > **Thanks for the comment**
> > >
> > > Thank you for your comment.
> > > Since both LazyTensor and Terra propose novel execution models of an imperative DL program, we think that both systems are great works and they are worth publishing at top-notch conferences.
> > > Both systems require significant engineering efforts, thus they require long time to develop.
> > > We hope that our decision not to upload our work to arXiv (and open-source our code) does not affect the decision of our submission.

---

> > > ### Author Response · Authors · 2021-09-06
> > > **Thanks for the comment**
> > >
> > > We would like to share some experiment results to compare LazyTensor and Terra.
> > > Since LazyTensor and Terra are built on different frameworks, PyTorch and TensorFlow, respectively, it is hard to perform an apple-to-apple comparison between LazyTensor and Terra directly.
> > > Therefore, we emulated LazyTensor on Terra by changing Terra's co-execution to lazy execution as follows.
> > > Before the `PythonRunner` reaches a tensor materialization point of a program, the `GraphRunner` waits for the execution.
> > > When the `PythonRunner` needs to materialize a tensor, the `PythonRunner` invokes the execution of the `GraphRunner` if the current trace is matched so far.
> > >
> > > We evaluated our LazyTensor emulation for ResNet50, BERT-Q&A, and DCGAN, which are simple programs to convert among our benchmarks.
> > > These programs do not entail a dynamic control flow, and the shape of intermediate tensors can be fully inferred without redundant tensor materializations.
> > >
> > > The following table denotes the training speed up of Terra, AutoGraph, and LazyTensor emulation relative to TensorFlow imperative execution.
> > >
> > > | DL Program        | Terra               | AutoGraph           | LazyTensor Emulation     |
> > > |:------------------|:-------------------:|:-------------------:|:------------------------:|
> > > | ResNet50          | x1.25 | x1.25 | x1.13 |
> > > | BERT Q&A          | x1.20 | x1.26 | x0.94 |
> > > | DCGAN             | x1.56 | x1.57 | x1.34 |

---

> > > > ### Comment · Reviewer_4LTS · 2021-09-08
> > > > **Thanks to the authors for the experiments**
> > > >
> > > > Thanks to the authors for the quick turnaround on the experiments - I realize it's a significant amount of work, especially during the rebuttal period.
> > > >
> > > > Do you have any sense of what's causing the gap in performance between your LazyTensor emulation and Terra? It would be nice to see some numbers that reflect the differences in approach free from confounding factors like implementation optimizations. For example, how often LazyTensor needs to split the graph vs. Terra.
> > > >
> > > > On a semi-related note, I was thinking about it, and I realized that I don't quite understand how Terra works with XLA. Here's an example of a situation I'd like clarification on.
> > > >
> > > > ```
> > > > b = tf.matmul(a)
> > > > custom_value = a.numpy() * 2
> > > > c = tf.matmul(tf.tensor(custom_value))
> > > > return c
> > > > ```
> > > > Specifically, in this situation, there would be an `OutputFetch` on the `a.numpy()` call, and an `InputFeed` when creating `tf.tensor(custom_value)`. How is that lowered to an XLA graph? If I understand correctly, you must be splitting the XLA graph at such points, correct? More generally, I would like some more details in the paper on how the `InputFeed` and `OutputFetch` operations interact with lowering the symbolic graph to a static compiler like XLA. I apologize for initially breezing past this detail, but upon reflecting upon it more I realized that it's non-trivial.
> > > >
> > > > If I understand correctly, if in our symbolic graph, one of the `InputFeed` operations depend on `OutputFetch`, then we cannot execute the whole graph in one program. Thus, we must split the graph when this happens. But how can we know what `InputFeed` nodes depend on previous `OutputFetch` nodes?
> > > >
> > > > Regardless of these clarifications, I'd like to thank the authors for their hard work in running updated experiments, both in combination with XLA as well as in comparisons against a LazyTensor "ish" approach. I think this last clarification is my final reservation before raising score. If the authors could provide a more in-depth analysis of how Terra outperforms LazyTensor, that would be even better.

---

> > > > > ### Author Response · Authors · 2021-09-08
> > > > > **Thanks for the comment**
> > > > >
> > > > > We first clarify how Terra works with XLA and then present the analysis of why Terra outperforms the LazyTensor emulation.
> > > > >
> > > > > ### XLA Compilation
> > > > >
> > > > > For XLA optimization, we used **Auto-clustering** XLA compilation of TensorFlow.
> > > > > Auto-clustering traverses a TensorFlow graph, and clusters TensorFlow operations that can be compiled and executed by XLA.
> > > > > The clustered operations are represented as a single XLA launch operation in the graph.
> > > > > The operations that cannot be compiled and executed by XLA are not modified during clustering.
> > > > >
> > > > > Terra indirectly handles a data dependency between an `OutputFetching` operation and an `InputFeeding` operation.
> > > > > We add the control dependencies (defined in TensorFlow) between `OutputFecthing` operations that should be executed prior and an `InputFeeding` operation.
> > > > > Since the `TraceGraph` of Terra captures the execution orders between the collected operations, we can figure out control dependencies.
> > > > >
> > > > > ```
> > > > > b = tf.matmul(a)
> > > > > custom_value = a.numpy() * 2
> > > > > c = tf.matmul(tf.tensor(custom_value))
> > > > > return c
> > > > > ```
> > > > > When Terra executes the above code, Terra generates the following symbolic graph.
> > > > > ```
> > > > > [Op that produces a]   ---> [MatMul that produces b]
> > > > >                         \---> OutputFetching ···
> > > > >    ···> InputFeeding ---> [MatMul that produces c]
> > > > > ```
> > > > > The solid arrow denotes data dependency and the dotted arrow denotes control dependency between two operations.
> > > > > Note that `a.numpy() * 2` is not captured by the graph since the expression does not create any TensorFlow operation.
> > > > > The `PythonRunner` executes the statement.
> > > > >
> > > > > After applying XLA compilation with Auto-clustering, the graph becomes as follows.
> > > > > ```
> > > > > [XLA Kernel of 'Op that produces a' and 'MatMul that produces b'] ---
> > > > >    ---> OutputFetching ···> InputFeeding ---
> > > > >       ---> [XLA Kernel of 'MatMul that produces c']
> > > > > ```
> > > > > Currently, since `OutputFetching` and `InputFeeding` operations could not be clustered, the two `MatMul` operations are also not clustered in a single XLA kernel. Nevertheless, the outer TensorFlow symbolic graph is a single graph.
> > > > > If we implement the lowering rule of `InputFeeding` and `OutputFetching` operations to HLO IR, we could create a single XLA kernel in this example.
> > > > >
> > > > > ### Analysis of the Performance Difference between Terra and our LazyTensor Emulation
> > > > >
> > > > > There are two reasons why Terra outperforms our LazyTensor emulation.
> > > > > First, the main performance gap comes from the LazyTensor's sequential execution of the `PythonRunner` and the `GraphRunner`.
> > > > > Second, the graph splits incur overheads. Terra creates a single graph for the entire program. On the other hand, our emulation of LazyTensor creates at most two forward subgraphs for the three models we experimented with.
> > > > > The other programs in the benchmark would create more than two subgraphs because these models are much more complicated than the three models.

---

> > > > > > ### Comment · Reviewer_4LTS · 2021-09-09
> > > > > > **Discussion**
> > > > > >
> > > > > > Thanks for the clarification. So if I understand correctly, any time we have an InputFeed after an OutputFetch, you need to break the XLA graph, right? How would we create a single XLA kernel in that situation - it seems like we need some mechanism that allows XLA to redispatch to Python in the middle of an XLA kernel?
> > > > > >
> > > > > > Thanks for the preliminary analysis of Terra vs. LazyTensor. I think a more extensive analysis of the strategies would significantly improve a final version of the paper.
> > > > > >
> > > > > > Overall, thanks for the extensive discussion and new experiments. I think adding  the XLA results presented in the first round of discussion (with a discussion of how the InputFeed/OutputFetch operations interact with static graph compilers), as well as a proper comparison against LazyTensor (with a deeper analysis of LazyTensor stalls/splits) would substantially improve this paper.
> > > > > >
> > > > > > The ideas presented in this paper are quite interesting, and I think they will be valuable to the community.
> > > > > >
> > > > > > I'm glad to accept this paper now, and I'll raise my score to a 7, and I hope that the authors will take the suggestions into account for any future versions of the paper :)
> > > > > >
> > > > > > PS: One more question - why does Terra outperform Tensorflow eager when no XLA is involved? Is it due to other graph optimization systems like Grappler?

---

> > > > > > > ### Author Response · Authors · 2021-09-09
> > > > > > > **Discussion**
> > > > > > >
> > > > > > > We appreciate your positive feedback on our work! If this paper gets accepted, we will reflect the reviewers' comments.
> > > > > > >
> > > > > > > Yes. The current implementation of Terra breaks the XLA graph when there is an `InputFeeding` operation after an `OutputFetching` operation.
> > > > > > > To create a single XLA kernel in our example, we should lower `InputFeeding` and `OutputFetching` operations preserving their control dependencies.
> > > > > > > Moreover, we need to allow communication between the `PythonRunner` and XLA Runtime during the co-execution.
> > > > > > > In this case, XLA Runtime would become the `GraphRunner` of Terra.
> > > > > > >
> > > > > > > Finally, Terra outperforms TensorFlow Eager execution when no XLA is involved because of the optimization of Grappler.

---

### Official Review · Reviewer_UnHn · 2021-07-17

**Rating:** 5
**Confidence:** 4

**Summary:**

The paper describes a framework based on imperative symbolic co-execution to execute deep learning programs. The main idea is to build a symbolic trace of DL operators while executing imperatively until the current iteration's trace is fully captured in its trace graph. At this point the imperative code and the symbolic code is run asynchronously with necessary communication inserted to guarantee correctness. The results show improved performance compared to AutoGraph (a one time trace only system).

**Ethical Concerns:**

I do not see any major ethical concern arising from this work.

**Limitations And Societal Impact:**

I have listed the technical limitations in the above section. I do not see any major negative societal impact arising from this work.

**Main Review:**

# Strengths

* The system uses the concept of trace graph to gradually build a symbolic graph of multiple traces. These ideas are exploited in the JIT compilation and dynamic binary instrumentation field (Code Caches) and it is commendable it is being utilized in the context of a DL framework.
* Comparison against other tracing imperative systems such as Autograph

# Weaknesses

* The authors completely overlook statically compiled DL backends such as XLA or TVM. These build symbolic data flow graphs and statically compile them to machine code and exploits better optimizations compared to TensorRT and other runtimes. The authors do not compare the performance of their system against such compiled approaches, which is a significant omission. In Tensorflow, you can easily invoke the XLA compiler by using tf.function regions.
* Related work missing on compiled approaches such as XLA or TVM. Please add them. XLA is only mentioned as a citation for operator fusion, but in fact is a fully featured compiler that does many other optimizations.
* Evaluation mentions it outperforms AutoGraph by upto 1.61x. A breakdown of results per benchmark would give a wholistic picture to appreciate this result in full. It's surprising why a graph is not included for this result.

**Time Spent Reviewing:**

3

---

> ### Author Response · Authors · 2021-08-10
> **Response to Reviewer UnHn**
>
> We thank the reviewer for suggestions for improvement.
> We answer the questions below.
>
> Both XLA and TVM are additional optimization steps that can be applied to a symbolic graph.
> When a user builds a symbolic graph (e.g., TensorFlow Graph and TorchScript IR), the user can
> convert the graph to either HLO IR (XLA) or Relay IR (TVM) to exploit their additional optimizations.
> Thus, we think that our work is orthogonal to XLA and TVM. We will clearly describe them as applicable optimizations of symbolic execution in the paper.
>
> The following table shows the training speed up when applying XLA to both Terra and AutoGraph.
> We do not include TVM because Terra focuses on training and TVM supports only inference workloads.
>
> | DL Program        | Terra               | Terra + XLA         | AutoGraph           | AutoGraph + XLA     |
> |:------------------|:-------------------:|:-------------------:|:-------------------:|:-------------------:|
> | ResNet50          | x1.25 | x1.38 | x1.25 | x1.40 |
> | BERT Q&A          | x1.20 | x1.42 | x1.26 | x1.48 |
> | YOLOv3            | x1.61 | x1.54 | x1.67 | x1.65 |
> | DCGAN             | x1.56 | x1.61 | x1.57 | x1.68 |
> | GPT2              | x1.27 | N/A   | x1.29 | N/A   |
> | BERT CLS          | x1.06 | x1.33 | N/A   | N/A   |
> | DropBlock         | x1.25 | x1.38 | N/A   | N/A   |
> | SDPoint           | x1.22 | x1.27 | N/A   | N/A   |
> | FasterRCNN        | x1.14 | N/A   | N/A   | N/A   |
> | Music Transformer | x1.30 | x1.73 | N/A   | N/A   |
>
> As reported in Figure 5 of the paper, each number denotes the amount of speed up relative to TensorFlow imperative execution.
> The values of 'Terra' and 'AutoGraph' are the same as the ones reported in Figure 5.
> We will update Figure 5 to include the results of Terra+XLA and AutoGraph+XLA to make clear that Terra can take advantage of additional XLA optimizations.
>
> When we apply XLA to Terra and AutoGraph, XLA is not applicable (N/A) to GPT2 and FasterRCNN due to the dynamic shape of the input data.
> For each training iteration, the shapes of input data to the models can change.
> However, XLA assumes static shapes, so it does not efficiently handle dynamic shapes; it compiles kernels every time the shape changes.
>
> As you can see, both Terra and AutoGraph achieve higher performance when applying XLA compilation except for YOLOv3.
> We profile the execution and find that the current XLA fails to efficiently cluster operations for YOLOv3, thus fails to create good kernels in YOLOv3.
> To be more specific, YOLOv3 includes some computation operations such as `ResizeNearestNeighbor` and `Where`, which are not supported by XLA. So, XLA cannot efficiently fuse computation operations.
> Additionally, we observe that Terra's performance decreases more than that of AutoGraph because the schedules of some `OutputFetching` operations are reordered because of XLA kernels, causing a longer stall in the `PythonRunner`.
> However, this problem can be addressed by extending XLA to support our `OutputFetching` operation.
>
> For the last weakness point, we would like to clarify that Terra outperforms imperative execution by up to 1.61x, not AutoGraph (Line 76 in the paper). Terra and AutoGraph share the same upper bound of performance improvement from the symbolic execution of TensorFlow.
>
> In Figure 6, we show the performance breakdown of the `PythonRunner` and the `GraphRunner` of Terra. We'd appreciate it if we'd get more information about the breakdown that is good to report.

---

> > ### Author Response · Authors · 2021-09-09
> > **Discussion**
> >
> > Does our response address your concerns? If there is any further information you need, please let us know.

---

### Official Review · Reviewer_JQgH · 2021-07-20

**Rating:** 9
**Confidence:** 4

**Summary:**

This paper proposes a combined approach based on both imperative and symbolic co-execution  of a program that can handle any imperative Deep Learning program while achieving the performance of symbolic graph execution. They evaluated on real-world models for imagenet and demonstrate their improvement.

**Main Review:**

Originality:
- In the context of DL program context, combinations of automatically building the symbolic graph from program traces seems new and interesting. I actually learned a lot from the paper.

Quality:
- The lazy construction of the graph makes sense in this context.
- The evaluation is solid on 10 large models and the gain is significant.
- The only problem is there is no formal argument of the correctness of the approach. However, it intuitively makes sense.
- Authors also should consider providing ablation studies showing how much time is required to build the graph.

Clarity:  Well written paper.

Significance: This kind of optimization has the potential to significantly improve the DL models.



**Time Spent Reviewing:**

3 hours

---

> ### Author Response · Authors · 2021-08-10
> **Response to Reviewer JQgH**
>
> We thank the reviewer for the comments.
>
> Terra does not affect the correctness of an imperative DL program because Terra preserves the program's semantics.
>
> The following table shows the time for graph generation for each DL program.
>
> | DL Program           | Avg. Trace Time [ms]    | Graph Gen. Time [ms] |
> |:------------------|------------------------:|---------------------:|
> | ResNet50          | 747.05  | 317.86  |
> | BERT Q&A          | 1016.49 | 610.09  |
> | YOLOv3            | 2886.46 | 1690.87 |
> | DCGAN             | 138.79  | 84.52   |
> | GPT2              | 947.59  | 480.59  |
> | BERT CLS          | 1239.83 | 665.8   |
> | DropBlock         | 701.65  | 333.96  |
> | SDPoint           | 547.18  | 309.93  |
> | FasterRCNN        | 2421.81 | 1414.16 |
> | Music Transformer | 754.55  | 312.97  |
>
> 'Avg. Trace Time' denotes the time to create a TraceGraph by collecting and merging traces, and 'Graph Gen. Time' denotes the time to convert the TraceGraph into a symbolic graph. Table 2 in our supplementary material shows that the number of collected traces in our evaluation does not exceed 4. Thus, this graph generation time is a very small portion of training a DL model, so it is negligible.

---

### Decision · Program_Chairs · 2021-09-27

**Decision:**

Accept (Poster)

**Comment:**

The reviewers appreciated the novel techniques presented in the paper for co-executing the symbolic and imperative representations of a model program. During discussions, one major concern that came up was regarding experimental evaluation and comparisons to other related approaches. The additional experiments of applying XLA to Terra, and comparison with LazyTensor emulation helped clarify some of the advantages of Terra's approach and the corresponding benefits. It would be great if authors can incorporate the detailed feedback from reviews, the additional experiments and results, as well as discussions about limitations and clarifications from the author response in the final version of the paper.